# Molecular Pathogenic Mechanisms of Hypomyelinating Leukodystrophies (HLDs)

Tomohiro Torii [1,2,3,*] and Junji Yamauchi [1,4,*]

1 Laboratory of Molecular Neurology, Tokyo University of Pharmacy and Life Sciences, Hachioji 192-0392, Japan
2 Laboratory of Ion Channel Pathophysiology, Graduate School of Brain Science, Doshisha University, Kyotanabe-shi 610-0394, Japan
3 Center for Research in Neurodegenerative Disease, Doshisha University, Kyotanabe-shi 610-0394, Japan
4 Department of Pharmacology, National Research Institute for Child Health and Development, Setagaya-ku 157-8535, Japan
* Correspondence: ttorii@mail.doshisha.ac.jp (T.T.); yamauchi@toyaku.ac.jp (J.Y.);
Tel.: +81-774-65-6055 (T.T.); +81-42-676-7164 (J.Y.); Fax: +81-42-676-8841 (J.Y.)

**Abstract:** Hypomyelinating leukodystrophies (HLDs) represent a group of congenital rare diseases for which the responsible genes have been identified in recent studies. In this review, we briefly describe the genetic/molecular mechanisms underlying the pathogenesis of HLD and the normal cellular functions of the related genes and proteins. An increasing number of studies have reported genetic mutations that cause protein misfolding, protein dysfunction, and/or mislocalization associated with HLD. Insight into the mechanisms of these pathways can provide new findings for the clinical treatments of HLD.

**Keywords:** oligodendrocytes; myelin; hypomyelination; demyelination; hypomyelinating leukodystrophy (HLD)





## 1. Introduction

Oligodendrocytes form myelin, which controls the velocity of axonal impulse conduction and protects axons in the central nervous system (CNS) [1,2]. Recent findings have demonstrated that stress [3] and brain plasticity [1] are associated with oligodendrocyte differentiation and myelination. With the introduction of next-generation sequencing methods, multiple autosomal recessive hypomyelinating leukodystrophy (HLD) genes have been identified and characterized (https://www.omim.org, accessed on 9 July 2023). Online Mendelian Inheritance in Man (OMIM) is a comprehensive database that provides information on genes, genetic phenotypes, and the relevance between disease and responsible genes based on previous studies. The number of HLDs and responsible genes identified in the last 10 years has increased (OMIM ID in parentheses): HLD1 (312080): *PLP1*; HLD2 (608803): *GJC2/GJA2*; HLD3 (260600): *AIMP1*; HLD4 (612233): *HSPD1*; HLD5 (610532): *HYCC1/FAM126A*; HLD6 (612438): *TUBB4A*; HLD7 (607694): *POLR3A*; HLD8 (614381): *POLR3B*, HLD9 (616140): *RARS*; HLD10 (616420): *PYCR2*; HLD11 (616494): *POLR1C*; HLD12 (616683): *VPS11*; HLD13 (616881): *C11ORF73*/HIKESHI; HLD14 (617899): *UFM1*; HLD15 (617951): *EPRS*; HLD16 (617964): *TMEM106B*; HLD17 (618006): *AIMP2*; HLD18 (618404): *DEGS1*; HLD19 (618688): *TMEM63A*; HLD20 (619071): *CNP*; HLD21 (619310): *POLR3K*; HLD22 (619328): *CLDN11*; HLD23 (619688): *RNF220*; and HLD24 (619851): *ATP11A*. Other leukodystrophies, including Aicardi-Goutières syndrome (AGS) [4], Adrenoleukodystrophy (ALD) (mutation in *ABCD1* (ATP binding cassette subfamily D member 1)) [5], Alexander disease (mutation in glial fibrillary acidic protein (GFAP)) [6], Canavan disease (mutation in *aspartoacylase* gene) [7], Cerebrotendinous xanthomatosis (mutation in *CYP27A1* gene) [8], Krabbe disease/globoid cell leukodystrophy (*Galactocerebrosidase* gene deficiency) [9], metachromatic leukodystrophy [10], and

Niemann–Pick disease (*sphingomylelin phosphodiesterase 1* gene deficiency (Type A or B) or mutation in *NPC1* (*Niemann–Pick type C1*) or *NPC2* gene) [11], were also characterized, and responsible genes were identified, respectively. Patients with HLDs reveal congenital cerebral hypomyelination, and other leukodystrophies show other white matter defects, such as demyelination (loss of myelin) and abnormal myelination. HLDs and others are classified on this point. However, both patients show diffuse hyperintensities on T2-weighted magnetic resonance imaging (MRI).

Congenital HLDs are a rare group of disorders characterized by a myelin deficit of the brain that is identified using magnetic resonance imaging (MRI). Patients with HLDs typically have nystagmus and motor deficits. In most cases, symptoms begin in infancy and include problems with feeding, a whistling sound when breathing, progressive spasticity leading to joint deformities (contractures) that restrict movement, speech difficulties (dysarthria), ataxia, and seizures.

In this review, we describe the essential roles and abnormal functions of genes associated with HLD and their effects on cells and/or oligodendrocytes, respectively.

## 2. HLD1 (OMIM ID 312080)

Proteolipid protein 1 (PLP1) is abundantly expressed in mature myelinating oligodendrocytes. Based on genomic mutation analysis, gene duplications, missense mutations, point mutations, and deletions in the *PLP1* gene are known to cause Pelizaeus–Merzbacher disease (PMD) (Figure 1) [12–15]. At least 50 mutations in *PLP1* have been identified, and several distinct types of PMD are recognized, such as conatal (most severe type), transitional (intermediate form), classical (mild PMD), and X-linked severe spastic paraplegia 2 (SPG2), based on the type of *PLP1* gene mutation [14]. The molecular pathogenesis of PMD includes the overexpression of a PLP1 protein localized and accumulated in the endoplasmic reticulum (ER) [15] and late endosomes/lysosomes in oligodendrocytes [16] and also causes ER stress and the accumulation of cholesterol, which results in the subsequent mis-sorting/mistrafficking of proteins and raft components (Figure 2A). The molecular machinery of pathogenesis has been demonstrated using induced pluripotent stem cells isolated from patients with PMD [17]. PLP1-overexpressing transgenic (tg) mice (PLP1-tg mice) were generated and characterized, and heterozygotes exhibited demyelination [18,19]. Interestingly, recent studies have demonstrated that the suppression of PLP1 protein improves myelination in brain lesions and possibly symptoms in HLD model mice [20,21]. Moreover, specific inhibitors (U0126 and PD98058) that block activity of mitogen-activated protein kinase 1/2 (MEK1/2) and extracellular signal-regulated kinase (ERK) signaling improved differentiation from PLP1-overexpressing oligodendrocyte precursor cells (OPCs) to myelinating oligodendrocytes, although excessive PLP1 protein inhibited cell differentiation (12). A recent study demonstrated that the microtubule-binding protein Tau was characterized as a marker of mature myelinating oligodendrocytes, and the number of Tau-positive oligodendrocytes was upregulated in lesions in PLP1-tg mice. These new findings may indicate that newly generated OPCs during demyelination differentiated into non-myelinating Tau-positive oligodendrocytes in the lesion. Interestingly, the cuprizone-induced demyelination model mice with acute demyelination and multiple sclerosis (MS) did not reveal the existence of these abnormal oligodendrocytes in the damaged regions. However, there are no data on whether the abnormal events are associated with the inhibition of re-myelination or not.

| | OMIM | Gene | Possible molecular pathogenesis |
|---|---|---|---|
| HLD1 | 312080 | *PLP1* | ER stress/Apoptosis |
| HLD2 | 608803 | *GJC2/GJA2* | ER stress/Apoptosis |
| HLD3 | 260600 | *AIMP1* | Dysfunction of AIMP1 |
| HLD4 | 612233 | *HSPD1* | mitochondria dysfunction |
| HLD5 | 610532 | *HYCC1/FAM126A* | Dysfunction of FAM126A |
| HLD6 | 612438 | *TUBB4A* | microtubule accumulation |
| HLD7 | 607694 | *POLR3A* | Dysfunction of RNA polymerase |
| HLD8 | 614381 | *POLR3B* | Dysfunction of RNA polymerase |
| HLD9 | 616140 | *RARS* | Dysfunction of RARS |
| HLD10 | 616420 | *PYCR2* | Dysfunction of PYCR2 |
| HLD11 | 616494 | *POLR1C* | Dysfunction of RNA polymerase |
| HLD12 | 616683 | *VPS11* | Apoptosis |
| HLD13 | 616881 | *C11ORF73/HIKESHI* | Dysfunction of C11ORF73 |
| HLD14 | 617899 | *UFM1* | unknown |
| HLD15 | 617951 | *EPRS* | Dysfunction of EPRS? |
| HLD16 | 617964 | *TMEM106B* | |
| HLD17 | 618006 | *AIMP2* | Dysfunction of AIMP2? |
| HLD18 | 618404 | *DEGS1* | mitochondria dysfunction? |
| HLD19 | 618688 | *TMEM63A* | Dysfunction of Autophagy? |
| HLD20 | 619071 | *CNP* | unknown |
| HLD21 | 619310 | *POLR3K* | Dysfunction of RNA polymerase |
| HLD22 | 619328 | *CLDN11* | unknown |
| HLD23 | 619688 | *RNF220* | unknown |
| HLD24 | 619851 | *ATP11A* | Abnormal PS and PC translocation |

**Figure 1.** Brief characteristics of HLDs. Responsible gene, OMIM ID for HLDs, and possible molecular pathology are represented, respectively. A multi-tRNA synthetase complex (MSC) consists of AIMP1, RARS, EPRS, and AIMP2. POLR3A, POLR3B, POLR1C, and POLR3K form the complex of RNA polymerase. Associations of these proteins are indicated with black line or blue line.

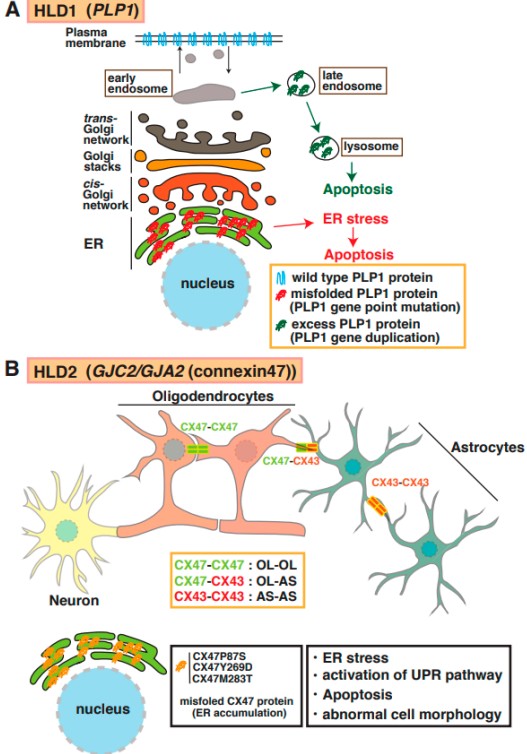

**Figure 2.** Molecular mechanisms underlying pathogenesis of HLD1 and HLD2. (**A**) PLP1 mutant and high-dose PLP1 protein cause hypomyelination and/or demyelination. Misfolded PLP1 protein (red) mainly accumulates at the endoplasmic reticulum (ER) and leads to ER stress and apoptosis. Excess PLP1 protein localizes in late endosomes and lysosomes. These events in oligodendrocytes are associated with subsequent demyelination. (**B**) Complexes of connexin (CX)47–CX47, CX47–CX43, or CX43–CX43 form functional gap junctions in oligodendrocytes (OL) and/or astrocytes (AS). Mutation in CX47 causes CX47 mislocalization and dysfunction of connexin-mediated glia–glia that leads to ER stress, activation of unfolded protein response (UPR) pathway, and abnormal cell morphology.

### 3. HLD2 (OMIM ID 608803)

Mutations of the gene encoding the gap junction protein connexin 47 (*Cx47*, also known as *GJA12* or *GJC2*) cause HLD2. As shown in Figures 1 and 2B, this protein is highly expressed in oligodendrocytes and forms a complex with itself (Cx47–Cx47) to control gap junction communication (GJC) between oligodendrocytes, and it associates with Cx43 (Cx47–Cx43) to communicate between oligodendrocytes and astrocytes [22]. The electrical and metabolic intracellular communications via gap junctions are responsible for the physiological function of oligodendrocytes. Some point mutations in *Cx47* (e.g., p.Ile33Met, p.Ser48Leu, p.Pro87Ser, p.Pro90Ser, p.Arg260Cys, p.Glu263Lys, p.Tyr272Asp, p.Met286Thr, and p.Met283Thr), and a frameshift mutation in *Cx47* [23,24] have been identified in patients with HLD2, who often have nystagmus, spastic ataxia, and hypomyelination/demyelination. Some of these mutations have been characterized; for example, the p.Pro87Ser, p.Tyr269Asp, and p.Met283Thr mutations of *Cx47* causes the protein to accumulate in the ER [25], and the p.Ile33Met mutation forms gap junctions but not functional homotypic channels [26]. The abnormal localization and non-functional Cx47 arising from these mutations lead to ER stress, the activation of the UPR pathway, apoptosis, and abnormal morphology that may be associated with hypomyelination and demyelination in patients with HLD2 (Figure 2B).

### 4. HLD3 (OMIM ID 260600) and HLD17 (OMIM ID 618006)

A few case reports have described mutations of the aminoacyl transfer RNA synthetase complex-interacting multifunctional protein-1 (AIMP1)/p43 that cause HLD3 (Figure 1) [27–29]. The AIMP1 protein forms a multi-tRNA synthetase complex (MSC) containing arginyl-tRNA synthetase (ArgRS) and glutaminyl-tRNA synthase, which are responsible for the translation and non-translation functions in the signaling of cell growth, immune response [30], and the survival of organisms (Figure 3a). In cancer cells, AIMP2 (Figure 3b) and KRS (Figure 3d) are phosphorylated and dissociated from MSC and subsequently control the inner cellular signaling transduction, including MAP kinase and cell adhesion molecules, respectively. Since the destabilization of this complex leads to a loss in enzyme activation of AIMP1 [29], MSC does not work properly to control protein synthesis.

A 2-bp deletion at position 292 (292CA) and a point mutation (Gln39 to Ter [p.Gln39Ter: Q39X]) results in a frameshift mutation and abnormal termination of protein synthesis, respectively. A recent study demonstrated that these AIMP1 mutants (Q39X and 292CA) mainly accumulate in the lysosome of neuronal cells and lead to the inhibition of the neuronal differentiation of the neuroblastoma cell line (Figure 3c) [28]. Also, the AIMP1 mutant specifically interacts with beta-actin, but not with wild type and AIMP1 Q39X mutant, in the lysosome and inhibits actin polymerization (Figure 3d). Thus, the AIMP1 mutant might suppress myelin formation via the actin cytoskeleton. Interestingly, ibuprofen, which is a nonsteroidal anti-inflammatory drug, is able to rescue the abnormal phenomenon. However, the relationship between the functional loss and accumulation of AIMP1 in lysosomes remains unclear.

A mutation in AIMP2 [p.Tyr35Ter (Tyr35 to Ter)] was identified in a patient with HLD17 who exhibited a progressive neurodevelopmental disorder [31]. AIMP2 mutant proteins resulted in mislocalization and aggregation in Golgi bodies in the FBD-102b oligodendrocyte cell line [32]. Since previous reports demonstrated that aggregated AIMP2 can interact with α-synuclein and facilitate α-synuclein aggregation in Parkinson's disease (PD) patients [31,32], aggregated AIMP2 might bind to proteins that regulate oligodendrocyte development, myelination, or molecular pathogenesis in patients with HLD17. Moreover, the complex, including AIMP1/2/3, is associated with multi-functional proteins such as PKC/NF-κB, p53, and JAK-STAT signaling pathways in various physiology, which are similar to those in cancer cells. Based on these findings, the disruption of AIMP2 function may affect these signaling transduction pathways in oligodendrocytes and/or neuronal.

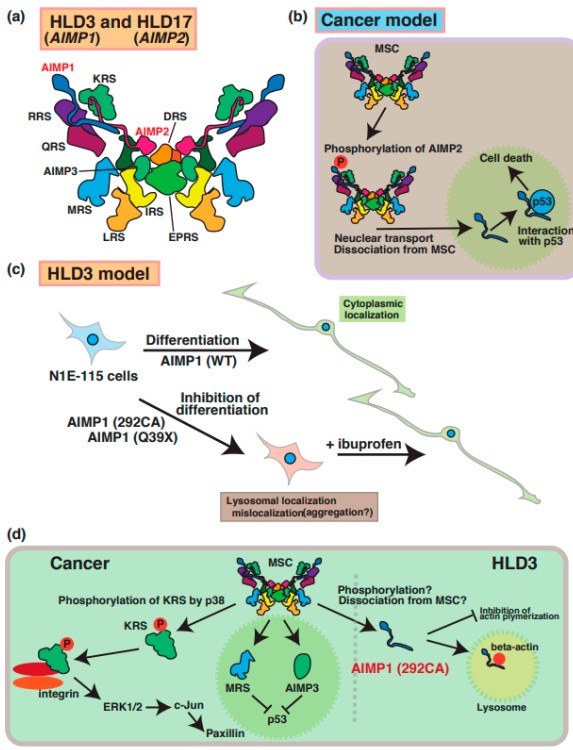

**Figure 3.** Molecular mechanisms underlying pathogenesis of HLD3 and HLD17. Aminoacyl tRNA synthetase complex interacting multifunctional protein 1 (AIMP1), AIMP2, and AIMP3 is included in the aminoacyl transfer RNA synthetase complex, as shown in panel (**a**). (**b**) In cancer cells, AIMP2 is phosphorylated and dissociated from MSC. Subsequently, phosphorylated AIMP2 is translocated from cytoplasmic to nucleus and associates with p53. (**c**) As a possible HLD3 pathology model, inhibitory effect of AIMP1 mutant in N1E-115 neuroblastoma cell lines. Both AIMP1 (292CA) and AIMP1 (Q39X) mutants localize in lysosomes and suppress differentiation of N1E-115 cells. Ibuprofen is able to rescue the phonotypes in AIMPI mutant-expressing cells. (**d**) An MSC member KRS is phosphorylated via p38 and regulates cell adhesion and/or cell migration via ERK1/2, c-Jun, and paxillin signaling pathways. Also, dissociated MRS and AIPM3 are translocated into nucleus and inhibit p53-mediated signal transduction. As a possible HLD3 pathology model, abnormal localization of AIMP1 292CA mutant is observed in lysosome and interacts with beta-actin. Also, the mutant, not wild type, inhibits actin polymerization. The abbreviations used are DRS, aspartyl-tRNA synthetase; EPRS, glutamyl-prolyl-tRNA synthetase; IRS, isoleucyl-tRNA synthetase; KRS, lysyl-tRNA synthetase; LRS, leucyl-tRNA synthetase; MRS, methionyl-tRNA synthetase; QRS, glutaminyl-tRNA synthetase; RRS; arginyl-tRNA synthetase, MSC; multi tRNA synthetase complex.

## 5. HLD4 (OMIM ID 612233)

Magen et al. first reported the identification of a point mutation in mitochondrial heat-shock protein 60 (Hsp60) in patients with HLD4 (Figure 1) [33] who showed hypotonia, nystagmus, and psychomotor development at the age of 3 months. Hsp60 (also called HSPD1), which is a molecular chaperone, forms a complex with Hsp10 and acts as a regulator for protein folding and trafficking inside mitochondria [34] (Figure 4A(a–c)). This study reported that the Hsp60 D29G mutant on chaperonin activity in *E. coli* decreased compared with wild-type Hsp60. Hsp60 also localizes in the cytosol and is associated with apoptosis signaling in cancer cells (Figure 4A(a)). The Hsp60 D29G mutant led to short mitochondria, and the motility of mitochondria, as well as the frequency of mitochondria fission and fusion, was dramatically decreased in Hsp60 D29G mutant-expressing cells (Figure 4A(b)) [35]. These abnormal morphological changes disrupt the essential functions of mitochondria, such as ATP production, that are necessary for enzyme activity and motor

protein-dependent protein transport. There is no evidence that the defect of Hsp60 activity is associated with mitochondria dysfunction and abnormal mitochondria morphology.

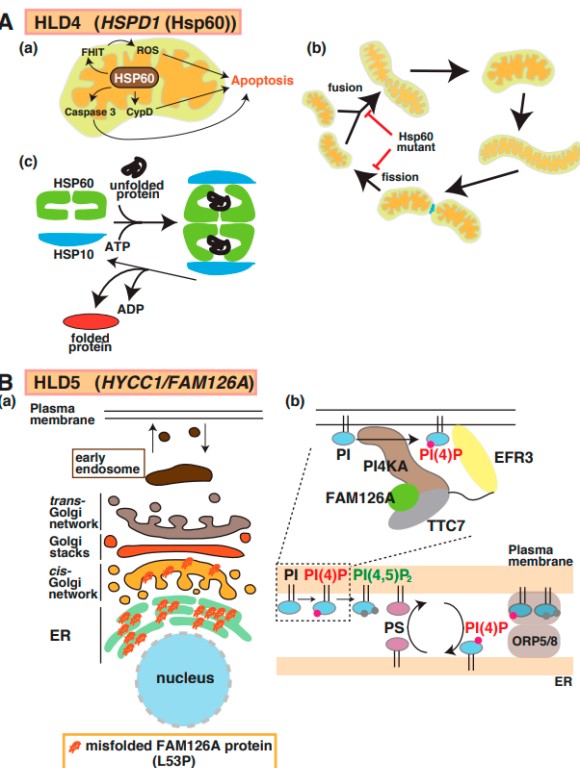

**Figure 4.** A possible molecular mechanism underlying HLD4 and HLD5 pathogenesis. (**A**) (**a**) Heat shock protein 60 (HSP60) controls apoptosis mediated via signal pathways, including caspase-3, cyclophilin D (CypD), and FHIT/ROS (top). (**b**) Hsp60 plays an important role in protein folding as chaperone (**c**). Hsp60 mutant inhibits mitochondria fusion and fission (bottom). (**B**) (**a**) The misfolded *FAM126A* (L53P) mutant accumulates in the endoplasmic reticulum (ER). (**b**) The complex, including PI4KA, EFR3, FAM126A, and TTC7, is illustrated, along with a key regulator to generate PI(4)P. FAM126A is responsible for synthesis of phosphatidylinositol 4-phosphate (PI(4)P), which controls some protein activity. Major myelin components PS and PI(4)P are exchanged in ORP5/8-mediated contact region between plasma membrane and ER. The abbreviations used are PS, phosphatidylserine; PI4KA, Phosphatidylinositol 4-Kinase Alpha; ORP5/8, oxysterol-binding protein (OSBP)-related protein.

## 6. HLD5 (OMIM ID 610532)

A mutation in the *Hyccin/FAM126A/DRCTNNB1A* gene is responsible for hypomyelination in HLD5 (Figure 1) [36]. FAM126A interacts with phosphatidylinositol-4 kinase 3 alpha (PI4K3alpha) and is responsible for the synthesis of phosphatidylinositol 4-phosphate (PI(4)P or PtdIns(4)P)) at the plasma membrane; it forms a complex, including PI4K3alpha and TTC7B, for recruitment to the plasma membrane (Figure 4B(b)) [37]. In addition, recent studies demonstrated that the complex containing oxysterol-binding protein (OSBP)-related protein 5 (ORP5) and ORP8 regulate contact between ER and plasma membrane. In the region, PI(4)P and phosphatidylserine (PS) are exchanged, as shown in Figure 4B(b). Because oligodendrocyte myelin consists of these lipids that regulate important events [38,39], the dysfunction of FAM126A possibly affects oligodendrocyte myelination mediated via lipid synthesis and lipid-associated signal transduction. Although wild-type FAM126A mainly localizes in the cytoplasm of cells, the accumulation of FAM126 L53P mutant in the ER has been observed in vitro and in vivo, as shown in Figure 4B(a) [12,40]. Transgenic mice expressing the myelin basic protein (MBP) promoter-driven FAM126 L53P mutant were generated as HLD5 model mice and analyzed. Interestingly, this mouse model indicated

significantly decreased expression of MBP in the corpus callosum and possibly abnormal myelination [12].

### 7. HLD6 (OMIM ID 612438)

Point mutations in the *TUBB4a* gene, which encodes a member of the beta-tubulin family that is brain specific and/or highly expressed in the brain, lead to hypomyelination with atrophy of basal ganglia and cerebellum (H-ABC [OMIM ID 612438]: p.Asp249Asn) [41], dystonia (DYT4 [OMIM ID 128101]: p.Arg2Gly or p.Ala271Thr) [42], isolated hypomyelination (p.Val255Ile and p.Arg282Pro) [41], and early infantile encephalopathy (p.Asn414Lys) [43]. A point mutation (p.Ala302Thr) [43] in *Tubb4a* was identified in the taiep rat, which revealed hypomyelination. Because the accumulation of microtubules was observed in oligodendrocytes of the rats and Tau was upregulated in cultured oligodendrocytes derived from the rats, abnormal microtubule formation and microtubule disorganization in oligodendrocytes may be included in the molecular pathogenesis of hypomyelination (Figure 1) [44].

A recent study identified novel *TUBB4A* mutants, including p.Asp295Asn, p.Arg46Met, p.Gln424His, and p.Arg121Trp, in patients with dystonia from four families; via in silico analysis, the function of these mutants was similar to that of the p.Arg2Gly mutant [45]. In contrast, the *TUBB4A* mutants p.Arg2Gly and p.Ala271Thr that cause dystonia 4 (DYT4) suppressed neurite extension in neuroblastoma and disorganized microtubule network in cells. Interestingly, these *TUBB4A* mutants affected mitochondrial transport [46] and interaction between tubulins and motor protein kinesin superfamily 5 (KIF5). Disruption of neuronal function in *TUBB4A* mutant-expressing cells may be associated with the pathogenesis of DYT4 (Figure 5A).

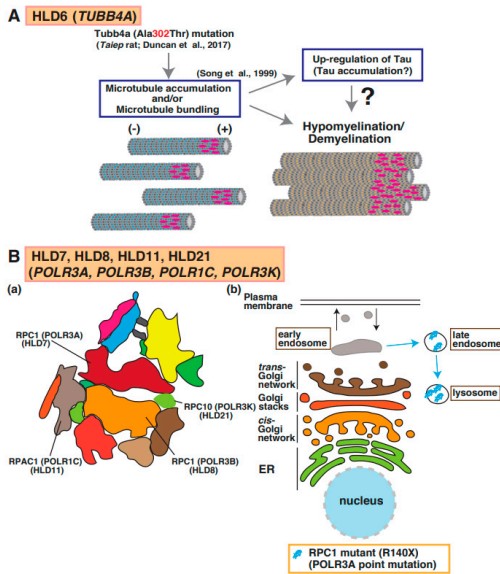

**Figure 5.** A possible molecular mechanism underlying HLD6, HLD7, HLD11, and HLD21 pathogenesis. (**A**) A mutation in Tubb4a (p.Ala302Thr) [43] was identified in demyelination model rats (*Taiep* rat). The rats show accumulation of microtubule and upregulation of microtubule binding protein Tau [44], which is expressed in mature myelinating oligodendrocytes. (**B**) (**a**) RNA polymerase subunits, including RPC1, RPC2, RPAC1, and RPC10, are illustrated. (**b**) A mutated *POLR3A* encoding RPC1 is localized in lysosomes, although wild-type RPC1 exists in the cytoplasmic region of oligodendrocytes.

### 8. HLD7 (OMIM ID 607694), HLD8 (OMIM ID 614381), HLD11 (OMIM ID 616494), and HLD21 (OMIM ID 619310)

The RNA polymerase (*POLR*)*3A*, *POLR3B*, *POLR1C*, and *POLR3K* genes encode DNA-directed RNA polymerase III subunits RPC1, RPC2, RPAC1, and RPC10, respectively,

and these proteins form a complex that regulates synthesis of 5S ribosomal RNA precursors, entire pools of transfer RNAs (tRNAs), U6 spliceosomal RNA, and a variety of other small RNAs [47]. RNA polymerase III forms a multi-subunit complex containing 17 subunits (Figure 5B(a)) [48]. Mutations in *POLR3A* cause HLD7 in patients and inhibit oligodendrocyte differentiation in vitro because the mutant abnormally accumulated in the lysosome (Figures 1 and 5B(b)) [49]. Patients have 4H syndrome (hypomyelination, hypodontia, and hypogonadotropic hypogonadism), ADDH (ataxia, delayed dentition, and hypomyelination), TACH (tremor-ataxia with central hypomyelination), or LO (leukodystrophy with oligodontia).

Also, mutations in the *POLR3B, POLR1C,* or *POLR3K* genes encoding RNA polymerase I and III subunit C cause HLD8, HLD11, or HLD21, respectively. It has been believed that the mutation-associated dysfunction (activity loss) of RNA polymerase III leads to tRNA deficiency and/or small molecule RNAs. However, the analysis for abnormal function of RNA polymerase III subunits to understand the pathogenesis of these diseases was still not performed.

## 9. HLD9 (OMIM ID 616140) and HLD15 (OMIM ID 617951)

Wolf et al. [50,51] and Nafisinia et al. [52] reported that point mutations (p.Asp2Gly, pCys32Trpfs*39, pArg512Gln, p.Asp2Gly, P.Ser456Leu, and p.Tyr616Leufs*6) in *RARS*, which encodes the cytoplasmic arginyl-tRNA synthetase (ArgRS), cause hypomyelination in patients with HLD9 (Figure 1). ArgRS constitutes a family of RNA-binding proteins that are responsible for the correct translation of the genetic code by covalently linking the appropriate amino acids to the 3′ end of the correct tRNA (Figure 6A(a)).

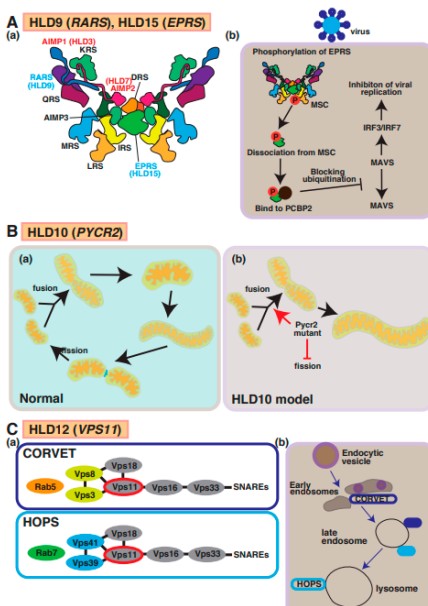

**Figure 6.** A possible molecular mechanism underlying HLD9, HLD15, HLD10, and HLD12 pathogenesis and normal functions. (**A**) (**a**) Multi tRNA synthetase complex (MSC) consists of RARS, EPRS, AIMP1, and AIMP2. (**b**) A signal transduction pathway of viral infection mediated via MSC. (**B**) (**a,b**) Pycr2 mutant inhibits fission and facilitates fusion, causing enlargement of mitochondria. (**C**) (**a**) Vps11 is a member of CORVET and HOPS complexes that regulate vesicle trafficking in cells. (**b**) Endocytic vesicle pathway is illustrated in CRVET-positive early endosomes, late endosomes containing CRVET and HOPS, and HOPS-positive lysosomes. The abbreviations used are DRS, aspartyl-tRNA synthetase; EPRS, glutamyl-prolyl-tRNA synthetase; IRS, isoleucyl-tRNA synthetase; KRS, lysyl-tRNA synthetase; LRS, leucyl-tRNA synthetase; MRS, methionyl-tRNA synthetase; QRS, glutaminyl-tRNA synthetase; RRS, arginyl-tRNA synthetase; MSC, multi tRNA synthetase complex; CORVET, class C core vacuole/endosome tethering; HOPS, homotypic fusion and protein sorting.

Mutations in the cytoplasmic glutamyl-prolyl-aminoacyl-tRNA synthetase (*EPRS*) gene (p.Arg339*, p.Pro1115Arg, p.Met1126Thr, p.Pro1160Ser, and p.Thr1223Leufs3*) were identified by Mendes et al. [53] and result in HLD15. EPRS catalyzes the aminoacylation of glutamic acid and proline tRNA. Some mutations of this gene cause decreased protein expressions and impaired aminoacylation enzyme activities by covalently linking an amino acid to its cognate tRNA during protein translation. A few studies have demonstrated these associations between gene mutation and hypomyelination, so the molecular mechanisms underlying the pathogenesis of HLD15 are largely unknown.

RARS and EPRS are members of MSC, which also consists of the HLD9- and HLD15-associated proteins AIMP1 and AIMP2, as described above (Figure 6(a)). Like AIMP1 and AIMP2, EPRS is phosphorylated and dissociated from MSC. Subsequently, phosphorylated EPRS binds to the host RNA-binding protein PCBP2, causing the inhibition of ubiquitination and viral replication via MAVS and IRF3/IRF7 in the viral infection signaling pathway (Figure 6A(b)). The molecular pathogenesis of HLD15 might be similar to this molecular mechanism. In another possible molecular mechanism of pathogenesis for HLD9, abnormal translation proteins responsible for oligodendrocyte myelination might cause hypomyelination in patients with HLD15.

## 10. HLD10 (OMIM ID 616420)

Mutations in the *PYCR2* gene encoding pyrroline-5-carcoxylate reductase family member 2, which is mainly localized in mitochondria and controls proline synthesis, have been detected in patients with HLD10 and hypomyelination (Figure 1). Several mutations in *PYCR2* were identified by Nakayama et al. (p.Arg119Ser and p.Arg251Cys) [54], Zaki et al. (p.Arg199Trp, p.Cys232Gly, and pArg266STOP) [55], and Meng et al. (p.Glu10Ter, p.Val193Met, and p.Arg266Ter) [56]. Both case reports have demonstrated that there are no significant changes in proline biosynthesis or other metabolites between normal subjects and patients with HLD10 that have a point mutation or *PYCR2* gene deficiency, indicating the *PYCR2* mutant might affect other signaling transduction pathways, not proline synthesis pathway. Interestingly, *PYCR2* deficiency leads to neurodegeneration mediated via the upregulation of Serine hydroxymethyltransferase 2 (SHMT2), associated with neurodevelopmental disorders with cardiomyopathy, spasticity and brain abnormalities, campylobacteriosis, and a subsequent high-level concentration of glycine in the brain [57].

On the other hand, a recent study reported that *PYCR2* mutants (p.Arg119Ser and p.Arg251Cys) result in the formation of large mitochondria due to increased fusion and decreased fission abilities in a mutant-expressing oligodendroglial cell line (FBD-102b) (Figure 6B) [58]. These cells revealed inhibiting oligodendroglial cell morphological differentiation, the hypofunction of *PYCR2* also affected mitochondria activities and may impair cellular signal transduction of oligodendrocyte differentiation and/or myelination. In mitochondria, it was believed that PYCR2 plays a role in the deubiquitination of target proteins as a component of the BRISC complex that is required for IFNAR1 deubiquitination in the BRISC complex [59]. Collectively, these reports indicated that PYCR2 is associated with signal transduction and proline synthesis.

## 11. HLD12 (OMIM ID 616683)

The tethering complexes homotypic fusion and protein sorting (HOPS) and class C core vacuole/endosome tethering (CRVET), including vascular protein sorting-associated protein 11 (Vps11), Vps18, Vps16, and Vps33, play roles in the endosomal/vacuolar transport. The complex associates with SNARE complexes and Rab GTPases (Rab5 and Rab7) and controls vesicle-mediated protein trafficking to the lysosome via the endocytosis and autophagic pathway (Figures 1 and 6C(a,b)) [60]. Vps11 is highly expressed in the platelet-derived growth factor receptor-alpha (PDGFRα)-positive OPCs and mature oligodendrocytes [60]. A demyelination mouse model (transgenic mice overexpressing *PLP1*) showed a lower expression of Vps11 [61]. A missense mutation, p.Cys846Gly, in Vps11, was first reported in a case study that described patients with a thin corpus callosum,

paucity of white matter, and delayed myelination [62]. Vps11 and Vps18 complexes are E3 ubiquitin ligases that control several signaling factors and pathways, including Wnt, estrogen receptor alpha (ERalpha), and NFκB. Because the mutation in Vps11 promotes ubiquitination-mediated protein degradation and impaired autophagy, the events result in oligodendrocyte cell death, the suppression of myelination, and possibly hypomyelination [63]. Another report demonstrated that the mutant inhibits oligodendrocyte differentiation via the downregulation of p70S6K signaling that controls the expression of myelin genes such as myelin regulatory factor (*MyRF*), *MBP*, and *PLP1* [64]. Collectively, these studies demonstrate that Vps11-dependent protein sorting mechanisms are responsible for oligodendrocyte myelination.

## 12. HLD13 (OMIM ID 616881)

Point mutations (p.Cys4Ser and p.Val54Leu) in the *HIKESHI* gene (also called *C11orf73*) were identified in patients with HLD13 (Figure 1) [65]. Heat-shock-induced nuclear import of 70 kDa heat-shock protein (Hsp70) is generally stress-inducible as it plays an important cytoprotective role in cells exposed to stressful conditions. Hikeshi was characterized as a transporter of Hsp70 from cytoplasm to nuclear to protect from heat-shock stress (Figure 7A(a,b)). A *C11orf73* mutation (Cys4 to Ser [C4S]) suppresses the differentiation of oligodendroglial FBD-102b cells [66]. Interestingly, a report has shown that Hsp70 specifically interacts with PLP1 and MBP in damaged oligodendrocytes in patients with multiple sclerosis [67], and the C11orf7 C4S mutant also specifically binds to filamin A. Considering these findings, the dysfunction of the *C11orf73* mutant might cause the mislocalization of MBP and PLP1 and abnormal cell morphological changes in oligodendrocytes in patients with HLD13 during development.

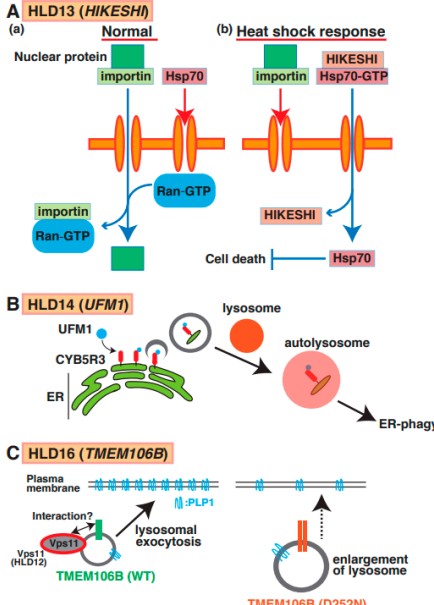

**Figure 7.** A possible molecular mechanism underlying HLD13, HLD14, and HLD16 pathogenesis and normal functions. (**A**) (**a**) Nuclear transport through nucleus pore is regulated via importin and GTP-form Ran in normal cells. On the other hand, Hsp70 does not go through nucleus pore. (**b**) Heat shock response suppresses importin-mediated nuclear transport and facilitates the interaction between HIKESHI and GTP-form Hsp70. And translocated Hsp70 is dissociated from the complex and inhibits cell death. (**B**) A molecular mechanism of ER-phagy mediated via autophagosome formation and fusion with a lysosome. CYB5R3 is UFMylated in surface of ER and controls autophagosome formation as initial stage of this event. (**C**) TMEM106B wild type mainly localizes in lysosome and controls lysosomal exocytosis to transport membrane protein such as PLP protein in oligodendrocytes. A study suggested that HLD12-associated protein Vps11 might interact with TMEM106B. TMEM106B D252N mutant expressing cells revealed lysosome enlargement, causing inhibition of PLP1 transport.

### 13. HLD14 (OMIM ID 617899)

A mutation (Arg81Cys) in the ubiquitin-like modifier (*UFM1*) gene, which is a post-translational modifier, was identified in patients with HLD14 (Figure 1) [68] and might lead to hypomyelination or impaired brain development. Ribosomal protein L26 (RPL26), elf4F complex, and histone H4 are UFMylated to control ER phagy, protein translation, and DNA damage response, respectively [69]. One study demonstrated that ER protein CYB5R3 was UFMylated and controlled ER phagy via the formation of autophagosomes and fusion with the lysosome (Figure 7B). However, un-UFMylated CYB5R3 did not act to impair brain development [70]. No reports describe which proteins are UFMylated or un-UFMylated in normal and damaged oligodendrocytes. Moreover, the associations between UFM1-dependent ER phagy and myelination have yet to be investigated.

### 14. HLD16 (OMIM ID 617964)

A mutation in *TMEM106B* (p.Asp252Asn) is associated with hypomyelination as a part of the clinical phenotype of HLD16 (Figure 1) [71]. TMEM106B is a type II transmembrane protein that is highly expressed and localized in the lysosome in oligodendrocytes and neurons [72]. Loss of TMEM106B affects gene expression patterns, including CNP, Olig2, and SOX10, the number of olig2-positive cells, oligodendrocyte differentiation, and trafficking of PLP1. No significant observations of myelin thickness and CNP/MBP protein level were observed in *TMEM106B* KO mice compared with wild-type mice [73]. Moreover, wild-type TMEM106B mainly localizes in lysosomes and controls lysosomal exocytosis to transport membrane proteins such as PLP protein in oligodendrocytes. TMEM106B D252N mutant-expressing cells revealed lysosome enlargement, causing the inhibition of PLP1 distribution (Figure 7C).

Some studies have demonstrated that three single-nucleotide polymorphisms (SNPs) in *TMEM106B* (risk allele) were identified as a risk factor for frontotemporal lobar degeneration (FTLD)-TDP in patients with the disease and resulted in the upregulation of *TMEM106B* mRNA expression in brain, suggesting that TMEM106B is a factor in the molecular pathogenesis of FTLD-TDP [74]. However, the functions of TMEM106B in oligodendrocytes have not yet been characterized.

### 15. HLD18 (OMIM ID 618404)

Mutations in delta 4-desaturase, sphingolipid 1 (*DEGS1*), which controls the sphingolipid metabolism pathway, have been identified in patients with HLD18. *DEGS1* is highly expressed in the CNS, localizes in the ER, and mediates the synthesis pathway from dihydroceramide to ceramide (Figure 1). A point mutation in *DEGS1* (p.Ala280Val) was identified, and the mutant protein was characterized by Karsai et al. [75]. The mutant caused impairment of the sphingolipid metabolic pathway, resulting in the upregulation of dihydroceramide in cells (Figure 8A(a,b)). Because sphingolipids are components of cellular membranes that are necessary for myelin sheath formation in oligodendrocytes, the impairment of the sphingolipid metabolic pathway might be associated with hypomyelination. A recent study demonstrated that *DEGS1* mutant-expressing cells exhibited the impairment of mitochondrial dynamics [76]. These findings suggest that the inactivation of mitochondria might be involved in the pathogenesis of HLD18.

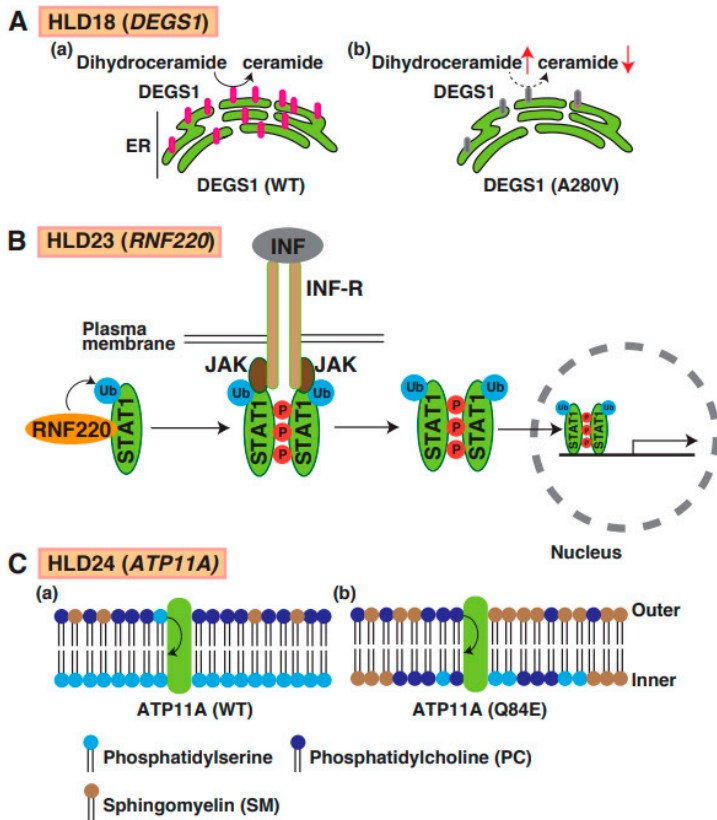

**Figure 8.** A possible molecular mechanism underlying HLD18, HLD23, and HLD24 pathogenesis and normal functions. (**A**) DEGS1-mediated ceramide synthesis mechanism in ER (**a**). The expression level of DEGS1 mutant (A280V) was dramatically decreased and caused upregulation of dihydroceramide and downregulation of ceramide (**b**). (**B**) In IFN-STAT1 signaling, RNF220 controls ubiquitination of STAT1, enhances phosphorylation of STAT1, and forms the complex, including JAK1 and INF receptors. Subsequently, ubiquitinated and phosphorylated-STAT1 is translocated into nuclear and controls transcription and subsequent protein synthesis. (**C**) (**a**) ATP11A regulates translocation of phosphatidylserine from outer plasma membrane to inner membrane phosphatidylcholine, but not phosphatidylcholine. (**b**) In PTP11A Q84E mutant expressing cells, distribution of lipids in membrane was changed because ATP11 mutant activity was dramatically decreased. Phosphatidylserine (PS), phosphatidylcholine (PC), and sphingomyelin (SM) are illustrated, respectively.

## 16. HLD19 (OMIM ID 618688)

Transmembrane protein 63A (*TMEM63A*) is a member of the mechanosensitive ion channel family, which is conserved from plants to humans. Fukumura et al. [77] reported that mutations in *TMEM63A* (p.Gly168Glu, p.Ile462Asn, p.Gly553Val, p.Tyr559His, and p.Gly567Ser) were identified in patients with HLD19 (Figure 1). A recent study demonstrated that TMEM63 specifically interacts with poly-ubiquitinated Toll-interacting protein (TOLLIP), which forms a complex with LC3 and controls autophagy in non-oligodendrocytes [78]; however, it is not known whether a mutation in *TMEM63A* affects autophagy. Since another mechanosensitive ion channel protein, Piezo1, controls myelination in Schwann cells [79], TMEM63A might be associated with myelination in oligodendrocytes.

## 17. HLD20 (OMIM ID 619071)

2′,3′-Cyclic nucleotide 3′-phosphodiesterase (CNP) is abundantly expressed in oligodendrocytes and is a well-known oligodendrocyte cell lineage marker for mature and myelinating oligodendrocytes. CNP also controls microtubule assembly and outgrowth in oligodendrocytes mediated via the interaction between CNP and tubulin during myelination [80].

A study reported a point mutation [p.Ser82Leu] in CNP, and the mutant-expressing cells showed abnormal actin organization (Figure 1) [81]. However, the molecular mechanisms of HLD20 pathogenesis are still unknown.

## 18. HLD22 (OMIM ID 619328)

Gow et al. generated *CLDN11* knockout (KO) mice, which showed an abnormal structure of a node of Ranvier and hindlimb weakness (Figure 1) [82]. It has been believed that oligodendrocyte-specific protein claudin-11 (CLDN11), a tight junction protein (TJ), is indispensable in the formation of the radial component of myelin and forms a diffusion barrier in myelin [83]. However, CLDN11 was also expressed in the basal cells like another claudin family since the mic also exhibited deafness and disappearance of TJs from the basal cells of the stria vascularis. Claudins consist of a PDZ-domain-binding motif that controls the direct interaction between claudins and cytoplasmic scaffolding proteins, including the TJ-associated proteins MUPP1, PATJ, ZO-1, ZO-2, ZO-3, and MAGUKs. In PNS, peripheral myelin protein 22 (PMP22), which is the main compartment of myelin in Schwann cells, associates with the TJ complex, including ZO-1 and occluding, to support the PMP22 function. These cell–cell interactions acting via cell adhesion molecules play a role in the regulation of intracellular signaling transductions. For example, CLDN11 enhances collective migration in squamous cell carcinoma mediated via the inhibition of small GTPases RhoA activity [84]. Some cell–cell adhesion molecules, including contactin, Necl1/4, and MAG, as well as CLDN11, are necessary for the myelination and control of the membrane–membrane contact of myelin. Some molecular mechanisms were characterized, as reported above, but oligodendrocyte-specific signaling transduction mediated via CLDN11 as a normal function is still being investigated.

Also, a recent study reported that the stop-loss variants in CLDN11 cause hypomyelinating leukodystrophy in patients with HLD22, who exhibit early-onset spastic movement disorders, expressive speech disorders, and eye abnormalities [85]. However, no study has yet described the molecular pathogenesis associated with the CLD11 signaling pathway.

## 19. HLD23 (OMIM ID 619688)

E3 ubiquitin-protein ligase RNF220 promotes the ubiquitination and proteasomal degradation of SIN3B [86], K63-linked polyubiquitination of STAT1 (Figure 8B) [87] and AMPA receptor [88]. Two homozygous missense variants in RNF220 [p.R363Q and p.R365Q] identified by Sferra et al. (Figure 1) [89] were associated with ataxia and deafness. A recent study reported that the number of cleaved caspase-3 positive cells did not change in Nestin-Cre; RNF220$^{fl/fl}$ conditional knockout mice (cKO) embryo compared with control mice, suggesting that RNF220-mediated signaling transduction is not responsible for cell proliferation and cell death. Interestingly, the number of OPCs in the hindbrain of RNF220 cKO embryo mice was also increased. However, it is unclear whether these abnormal events in damaged oligodendrocytes are associated with hypomyelination.

## 20. HLD24 (OMIM ID 619851)

The P4-ATPases ATP11A has flippase activity against phosphatidylserine and controls the translocation of phosphatidylserine from the outer to the inner leaflet of the plasma cell membrane (Figure 8C(a)). A recent study reported a mutation in the *ATP11A* gene (p.Gln84Glu) in a patient with HLD24 (Figure 1), and the expression of the mutant resulted in increased levels of sphingomyelin in the outer leaflet of the plasma membrane (Figure 8C(b)) [90]. The mutant abnormally translocated phosphatidylcholine as well as phosphatidylserine. *Atp11a* Q84E mutant heterozygous knock-in mice (*Atp11a*$^{Q84E/WT}$) were generated and revealed neurological defects; however, the specific functions of oligodendrocytes, including cell lineage and myelination, have not yet been analyzed.

## 21. Clinical Trial for Leukodystrophies

It has been reported that the safety and preliminary effectiveness of human central nervous system stem cells (HuCNS-SC) were transplanted into patients with HLD1 in a Phase 1 study since a previous study revealed that HuCNS-SC transplantation in well-established hypomyelination model mice (Shiverer) differentiated into oligodendrocytes and formed myelin that is confirmed using MRI [91]. PMD patients from 6 months to 5 years of age with point mutations in the *PLP1* gene were treated with HuCNS-SC transplantation. MRI and MR diffusion tensor imaging (DTI) can evaluate myelination in white matter after HuCNS-SC transplantation, indicating favorable results in most cases, and the treatments are safe [92,93]. Since reports about the HuCNS-SC transplantation are still few, follow-up reports are needed in international projects. Additionally, the knockdown of PLP1 using the CRIPSR-Cas9 system or RNA interference is also available for gene therapy as clinical treatment with plp1 gene duplication in PMD patients, as described above. Like HLD1, the overdose of PMP22 in Schwann cells of Charcot–Marie–Tooth 1A (CMT1A) patients causes demyelination in the peripheral nerve system (PNS). PXT3003 containing baclofen, naltrexone, and D-sorbitol was produced using genome data, and Artificial Intelligence (AI) technology reduced the expression of PMP22 and improved sensory function in CMT1A patients. Now, pharmaceutical companies are applying a similar strategy and trying to predict drug combinations for clinical treatment using developing AI technology as drug repositioning.

On the other hand, many researchers are also conducting RNA-seq analysis using next-generation sequencing (NGS) technology and have reported that some populations of non-myelinating and mature oligodendrocytes are identified and classified in wild-type mice and disease model mice. Since researchers are analyzing mRNA expressions in patients with HLDs based on these studies, these data also support the development of therapeutic agents.

## 22. Discussion and Conclusions

As described above, proteins encoding HLDs responsible genes control various signaling transduction. RNA synthesis-mediated *POLR3* family controls crucial roles in various cells. To clarify if the impairment of protein and RNA synthesis that were regulated via HLD-responsible gene products causes hypomyelination or not, researchers are still investigating. Interestingly, a recent study demonstrated that point mutation in the *POLR3B* gene ((p.Arg469Cys, p.Cys490Tyr, and p.Arg1046His)), which is a responsible gene for HLD9, leads to demyelinating Charcot–Marie–Tooth disease (CMT) phenotypes [94,95], suggesting inner cellular molecular pathological mechanism might be common between in CNS and PNS. These patients revealed demyelinating neuropathy and no additional neurological or extra-neurological involvement. Maple syrup urine disease (MSUD) is known as an autosomal-recessive disorder that impairs the metabolism pathway of branched-chain amino acids (BCAAs), including leucine, isoleucine, and valine in mitochondria. Also, patients with MSUD exhibit hypomyelination and/or dysmyelination. It has been believed that a deficiency or loss in enzyme activity of an enzyme complex branched-chain alpha-keto acid dehydrogenase, which is a key regulator of BCAA metabolism, subsequently leads to a high concentration of leucine and/or alpha-ketoisocaproic acid in the brain. The toxic composition accumulation causes neurological dysfunction and hypomyelination. The cellular function of BCAA was investigated, and some studies have reported BCAA controls the TCA cycle, mechanistic target of rapamycin (mTOR) signaling transduction, and amino acyl tRNA biosynthesis. It has been well characterized that mTOR plays a key role in oligodendrocyte myelination [96]. The onset of MS and HLD9 is regulated via mTOR signaling. The events might be associated with hypomyelination. Moreover, the deficiency of tRNA in MSUD patients may also cause hypomyelination, which is similar to HLDs (HLD7, 8, 9, 11, 15, and 21). However, little is known about the association between HLDs and different diseases such as MSUD.

Due to the development of next-generation sequencing technology and data science, accumulating studies have demonstrated that mutations in genes associated with different HLDs often cause hypomyelination and/or demyelination. However, the normal function and abnormal function of protein-encoding-HLD-responsible genes were still not characterized. When these molecular mechanisms are understood, these studies may contribute to establishing clinical treatments, including gene therapy and drug development. Wolf et al. have described a review of the HLDs and it clearly demonstrated pathogenic mechanisms of HLDs [97]. In this review, we upload information on HLDs onset from it based on recent reports that were published within past five years.

**Author Contributions:** Conceptualization, T.T. and J.Y.; writing—original draft preparation, T.T.; supervision, J.Y. All authors have read and agreed to the published version of the manuscript.

**Funding:** This research received no external funding.

**Institutional Review Board Statement:** Not applicable.

**Informed Consent Statement:** Not applicable.

**Data Availability Statement:** Not applicable.

**Conflicts of Interest:** The authors declare no conflict of interest.

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
