# Peer review of "Molecular Pathogenic Mechanisms of Hypomyelinating Leukodystrophies (HLDs)"

_2035-8377, doi:10.3390/neurolint15030072_

Round 1
Reviewer 1 Report (Previous Reviewer 2)
The manuscript is a review of the many HLD (hypodemyelinating disease subtypes) and as such provides a resource for finding literature relevant to the many diseases identified by alterations in a multitude of genes. The authors provide models indicating how possible defects in each of the genes may influence myelinating oligodendrocyte function and, hence, hypo- and/or demyelination. Some similar pathways, e.g., the AIMPs are covered in different sections, HLD3 and HLD17 (Figure 2) and HLD9, (Figure 5). Similarly, genes affecting mitochondria are also split. Consolidating these may help in understanding relationships among the HLD diseases.
The investigators, like many others, oversimplify the oligodendrocyte, failing to consider the complexity inherent in supplying constituents to many sheaths at the same time and to deploying organelles in processes and internodes to facilitate local control. For example, take ER stress. Is the ER that the authors are referring to perinuclear ER or does it include or specify ER that extends along process to and around myelinating internodes. The same can be said for mitochondria, plasma membrane and/or other intracellular organelles.
Nevertheless, this compilation will be a useful resource for those interested in understanding the many ways that defects in multiple proteins lead to common/related abnormalities and it is not my job to reorganize the review.
Author Response
The manuscript is a review of the many HLD (hypodemyelinating disease subtypes) and as such provides a resource for finding literature relevant to the many diseases identified by alterations in a multitude of genes. The authors provide models indicating how possible defects in each of the genes may influence myelinating oligodendrocyte function and, hence, hypo- and/or demyelination. Some similar pathways, e.g., the AIMPs are covered in different sections, HLD3 and HLD17 (Figure 2) and HLD9, (Figure 5). Similarly, genes affecting mitochondria are also split. Consolidating these may help in understanding relationships among the HLD diseases.
Response 1.
Thank you very much for your comments.
The investigators, like many others, oversimplify the oligodendrocyte, failing to consider the complexity inherent in supplying constituents to many sheaths at the same time and to deploying organelles in processes and internodes to facilitate local control. For example, take ER stress. Is the ER that the authors are referring to perinuclear ER or does it include or specify ER that extends along process to and around myelinating internodes. The same can be said for mitochondria, plasma membrane and/or other intracellular organelles.
Response 2.
Thank you very much for insightful comments. We hope this review help researchers to establish clinical treatment for HLDs.
Nevertheless, this compilation will be a useful resource for those interested in understanding the many ways that defects in multiple proteins lead to common/related abnormalities and it is not my job to reorganize the review.
Response 3.
Thank you very much for your comments.
Reviewer 2 Report (New Reviewer)
Your paper addresses a very complex but important and not frequently discussed subject on molecular and genetic mechanisms evoked for HLDs, particularly as an effort to concentrate so much material in one article.
(1) Many readers may not be familiar with genetic sources therefore the abbreviation in the introduction OMIM should be explained as 'Online Mendelian Inheritance in Man'. You already provide the ID numbers and the web address,
(2) The complexity and amount of material make the reading a little difficult at times particularly in the Discussion part. While the idea to divide the material into subsegments is welcome to assist clarity, the concentration of statements in the Discussion/Conclusion segment probably could be made clearer by using a table?
(3) Your figures are busy but of excellent illustration quality. This reviewer suggests using an example or examples of brain imaging of leukodystrophies, always representing clinical diagnostic challenges and management concern along with prognosis.
This reviewer feels a considerable English review and editing of the manuscript should be undertaken to make your concepts more comprehensible.
Author Response
(1) Many readers may not be familiar with genetic sources therefore the abbreviation in the introduction OMIM should be explained as 'Online Mendelian Inheritance in Man'. You already provide the ID numbers and the web address,
Response(1)
I apologize my mistake. According to reviewer's comment, I inserted explanation of OMIN into the sentence.
(2) The complexity and amount of material make the reading a little difficult at times particularly in the Discussion part. While the idea to divide the material into subsegments is welcome to assist clarity, the concentration of statements in the Discussion/Conclusion segment probably could be made clearer by using a table?
Response(2)
Thank you for important suggestion. I make a table to understand easily for the journal's readers.
(3) Your figures are busy but of excellent illustration quality. This reviewer suggests using an example or examples of brain imaging of leukodystrophies, always representing clinical diagnostic challenges and management concern along with prognosis.
Response(3)
Thank you for the constructive comments. It is very hard to represent brain imaging of HLD for us, since we do not belong to medical school. And it take long time to get permission from our university's Ethics Committee.
Round 2
Reviewer 2 Report (New Reviewer)
Thank you for your reply and for carrying out this reviewer's observations.
None
This manuscript is a resubmission of an earlier submission. The following is a list of the peer review reports and author responses from that submission.
Round 1
Reviewer 1 Report
The goal of the manuscript is to provide a summary of the hypomyelinating leukodystrophies. Although this could be an informative review, in its current state, the level of detail of the information that is provided is to superficial and incomplete to provide significant value to the reader.
Suggestions:
It is unclear how certain leukodystrophies are given an HLD classification and others (Krabbe disease/globoid cell leukodystrophy, metachromatic leukodystrophy, Alexander disease, X-linked adrenoleukodystrophy, Canavan disease, there may be others) are not. There may be a good explanation for how leukodystrophies are classified should be addressed in the beginning.
Also, there are some HLDs that are not presented: HLD 8, 11, and 21. Additionally, 17 is not discussed.
A bigger issue with the review is that it reads as a list of the HLDs and identifies the gene that is mutated but provides little to no information regarding each leukodystrophy. The review would be much more informative with a inclusion of histologic and clinical presentation and a discussion of current therapies, if there are any.
there is potential for such a review but much more detail for each disease should be presented.
Author Response
We would like to thank the reviewer’s for carefully reading our manuscript and for providing many constructive comments. Based on the reviewer’s comments, we have revised the entire manuscript accordingly.
Reviewer 1
The goal of the manuscript is to provide a summary of the hypomyelinating leukodystrophies. Although this could be an informative review, in its current state, the level of detail of the information that is provided is to superficial and incomplete to provide significant value to the reader.
Suggestions:
It is unclear how certain leukodystrophies are given an HLD classification and others (Krabbe disease/globoid cell leukodystrophy, metachromatic leukodystrophy, Alexander disease, X-linked adrenoleukodystrophy, Canavan disease, there may be others) are not. There may be a good explanation for how leukodystrophies are classified should be addressed in the beginning.
Response 1. Thank you for valuable comments. We have added sentence about classification of others leukodystrophy to introduction (Page 1, line 44) according to reviewer’s comment.
Also, there are some HLDs that are not presented: HLD 8, 11, and 21. Additionally, 17 is not discussed.
Response 2. Thank you for the constructive comments. We wrote information of HLD7, HLD8, HLD11, and HLD21 together, since proteins (RNA polymerase) encoding responsible genes form complex to regulate synthesis of 5S ribosomal RNA precursors, entire pools of transfer RNAs (tRNAs), U6 spliceosomal RNA, and a variety of other small RNAs. Also, we described and discussed possible molecular mechanisms of HLD17 pathogenesis (Page 4, line 117). Also, we inserted information of Hsp60 function and Figure 2C (Page 5, line 129 and line 134).
A bigger issue with the review is that it reads as a list of the HLDs and identifies the gene that is mutated but provides little to no information regarding each leukodystrophy. The review would be much more informative with a inclusion of histologic and clinical presentation and a discussion of current therapies, if there are any.
there is potential for such a review but much more detail for each disease should be presented.
Response 3. We understand the review is superficial, however, this is a first time to present information of newly identified HLDs based on our recent research. Of course, many researchers have identified mutations in responsible genes of HLDs but still super few studies have been reported HLD pathology except for HLD1 and also its normal function. We believe the review progress study and analysis for HLDs to understand especially molecular mechanisms (analysis of abnormal signal transduction) of pathogenesis.
Reviewer 2 Report
I believe that the authors have found an interesting area to review. Unfortunately, they did not do a good job. The abstract focuses on PLP1 and HLD1 and only in the last sentence is there a suggestion that the other HLD genes are discussed. In reading background literature to familiarize myself with the subject, I came across an important recent review (Wolf et al, Hypomyelinating neuropathies, Nature Reviews 2021) that wasn't cited. In reading the section on PLP1, I found the description was not logical or easy to follow. A recent reference (Khalaf , Mutations in proteolipi protein 1 (Biomedicine 2022) would have been useful to include. There is no attempt in the conclusion to try to tie any of the disorders together and having descriptions from so many it would have been nice to share thoughts as to why these different defects might lead to common/related diseases. The models shown in the figures are vague and not in all cases clearly described. For example, Figure 1B has nothing to do with the title of the Figure, Molecular mechanisms underlying HLD1 and HLD2. In Figure 2A, there is no information on the abbreviations: KRS, DRS, FHIT. Should readers have to look up the cited papers to understand the figures?
Although the English is readable, it is not always logical.
Author Response
We would like to thank the reviewer’s for carefully reading our manuscript and for providing many constructive comments. Based on the reviewer’s comments, we have revised the entire manuscript accordingly.
Reviewer 2
I believe that the authors have found an interesting area to review. Unfortunately, they did not do a good job. The abstract focuses on PLP1 and HLD1 and only in the last sentence is there a suggestion that the other HLD genes are discussed. In reading background literature to familiarize myself with the subject, I came across an important recent review (Wolf et al, Hypomyelinating neuropathies, Nature Reviews 2021) that wasn't cited. In reading the section on PLP1, I found the description was not logical or easy to follow. A recent reference (Khalaf , Mutations in proteolipi protein 1 (Biomedicine 2022) would have been useful to include. There is no attempt in the conclusion to try to tie any of the disorders together and having descriptions from so many it would have been nice to share thoughts as to why these different defects might lead to common/related diseases. The models shown in the figures are vague and not in all cases clearly described. For example, Figure 1B has nothing to do with the title of the Figure, Molecular mechanisms underlying HLD1 and HLD2. In Figure 2A, there is no information on the abbreviations: KRS, DRS, FHIT. Should readers have to look up the cited papers to understand the figures?
Response 1. Thank you for the constructive comments. According to reviewer’s comments, we have cited some reports that are described valuable information and support our review.
Response 2. We apologize for the insufficient explanation of Figures. We have modified the title of the Figure 1 and inserted abbreviations to figure legends (Figure 2) according to reviewer’s comments.
Response 3. We understand the review is superficial, however, this is a first time to present information of newly identified HLDs based on our recent research. Of course, many researchers have identified mutations in responsible genes of HLDs but still super few studies have been reported HLD pathology except for HLD1 and also its normal function. We believe the review progress study and analysis for HLDs to understand especially molecular mechanisms (analysis of abnormal signal transduction) of pathogenesis.
Round 2
Reviewer 1 Report
This is a significantly improved version of the manuscript. I recommend that the authors add more to the discussion of HLD10.
Author Response
Dear Reviewer 1,
Thank you for the constructive comment.
According to Reviewer 1 comment, we discussed about HLD10 and added the information in revised manuscript.
Reviewer 2 Report
The authors made minimal changes to the manuscript, where I had expected far more effort. They did not consider the references I suggested, nor did they make any attempts to link the different diseases, although others have done so. It is disappointing to indicate ways that the manuscript should be improved and have these ideas ignored.
The English is not the problem.
Author Response
Dear Reviewer 2,
We apologize for our previous revision. According to Reviewer's comments, we cited some references suggested by reviewer and discussed the association about HLDs and different disease.